# Impact of progressive resistance training on CT quantified muscle and adipose tissue compartments in pancreatic cancer patients

**Raoul Wochner**[1©]**, Dorothea Clauss**[2,3©]**, Johanna Nattenmüller**[1©]**, Christine Tjaden**[4]**, Thomas Bruckner**[5]**, Hans-Ulrich Kauczor**[1]**, Thilo Hackert**[4]**, Joachim Wiskemann**[3‡]**, Karen Steindorf**[2‡]*

**1** Department of Diagnostic and Interventional Radiology, Heidelberg University Hospital, Heidelberg, Germany, **2** Division of Physical Activity, Prevention and Cancer, German Cancer Research Center and National Center for Tumor Diseases, Heidelberg, Germany, **3** Division of Medical Oncology, National Center for Tumor Diseases and Heidelberg University Hospital, Heidelberg, Germany, **4** Department of General, Visceral and Transplantation Surgery, Heidelberg University Hospital, Heidelberg, Germany, **5** Institute for Medical Biometry and Computer Science, Heidelberg University Hospital, Heidelberg, Germany

© These authors contributed equally to this work.
‡ These authors also contributed equally to this work. These authors share last authorship on this work.
* k.steindorf@dkfz.de

**Data Availability Statement:** This study is based on human research participant data and was approved by the Ethics Committee of the Medical

## Abstract

### Objectives

Loss of body weight is often seen in pancreatic cancer and also predicts poor prognosis. Thus, maintaining muscle mass is an essential treatment goal. The primary aim was to investigate whether progressive resistance training impacts muscle and adipose tissue compartments. Furthermore, the effect of body composition on overall survival (OS) was investigated.

### Methods

In the randomized SUPPORT-study, 65 patients were assigned to 6-month resistance training (2x/week) or a usual care control group. As secondary endpoint, muscle strength of the upper and lower extremities was assessed before and after the intervention period. Routine CT scans were assessed on lumbar L3/4 level for quantification of total-fat-area, visceral-fat-area, subcutaneous-fat-area, intramuscular-fat-area, visceral-to-subcutaneous fat ratio (VFR), muscle-area (MA), muscle-density and skeletal-muscle-index (SMI). OS data were retrieved.

### Results

Of 65 patients, 53 had suitable CT scans at baseline and 28 completed the intervention period with suitable CT scans. There were no significant effects observed of resistance training on body composition ($p > 0.05$; effect sizes $\omega^2_p < 0.02$). Significant moderate to high correlations were found between MA and muscle strength parameters ($r = 0.57$–$0.85$; $p < 0.001$). High VFR at baseline was a predictor of poor OS (VFR$\geq$1.3 vs. <1.3; median OS 14.6 vs. 45.3 months; $p = 0.012$). Loss of muscle mass was also a predictor of poor OS (loss vs. gain of SMI; median OS 24.6 vs. 50.8 months; $p = 0.049$).

Faculty of the University of Heidelberg (S-409/2013). The patients' informed consent did not include public data sharing. Further, the small sample size of the study may facilitate the reidentification of patients even if we provide pseudonomized data only. Thus, some limits to open access are given. The non-author point of contact where data requests can be sent to is: openaccess@dkfz.de.

**Funding:** This work was supported by a grant of the Foundation German Cancer Aid (https://www.krebshilfe.de/; SUPPORT-Study Grant No. 110513 (KS, JW) and 110552 (TH) as well as by additional intramural financial support of the German Cancer Research Center and the University Hospital Heidelberg. The external funder had no role in study design, data collection and analysis, decision to publish, or preparation of the manuscript.

**Competing interests:** The authors have declared that no competing interests exist.

## Conclusion

There is anabolic potential in patients with resectable pancreatic cancer. A progressive resistance training may help patients to maintain their muscle mass and avoid muscle depletion. CT-quantified muscle mass at the level of L3/4 showed a good correlation to muscle strength. Therefore, maintaining muscle mass and muscle strength through structured resistance training could help patients to maintain their physical functioning. A high VFR at baseline and a high loss of muscle mass are predictors of poor OS. Registered on ClinicalTrials.gov (NCT01977066).

## Introduction

In many cancer patients, weight loss is frequently already present at the time of diagnosis [1]. Patients with pancreatic cancer in particular often suffer from severe weight loss and loss of muscle mass [2].

Pancreatic cancer is a frequent highly malignant disease with a very poor prognosis and consecutive high mortality with a 5-year survival rate across all stages of 6% [3]. Most pancreatic cancers are diagnosed at a late stage due to very late and unspecific symptoms [4]. In up to 74% of pancreatic cancer patients, cachexia, a multifactorial wasting syndrome characterized by an ongoing loss of muscle mass with or without the loss of fat mass, systemic inflammation and usually weight loss [5] is present [6]. Further, the loss of muscle mass and weight loss leads to reduced muscle strength which additionally worsens functional capacity. Besides functional impairments, patients with cachexia tend to have more fatigue and a poor prognosis [5, 7, 8]. The loss of muscle mass (MA) and body composition with a high ratio of visceral fat tissue to subcutaneous fat tissue (VFR) were reported as predictors of poor prognosis in patients with lung cancer [9]. Therefore, maintaining MA, physical functioning and quality of life are among the main treatment goals in pancreatic cancer patients.

Exercise is known to have positive effects on disease- and treatment-related side effects in cancer patients during and after cancer treatment such as improvements in physical fitness [10], quality of life [11] and fatigue [12]. Resistance training in particular is reported to have a positive effect on improving MA due to increased muscle protein synthesis and improving muscle metabolism [13]. Recently, our group showed that pancreatic cancer patients can benefit from progressive resistance training with regard to muscle strength and quality of life as part of the SUPPORT-study [14, 15]. First evidence also suggests that exercise plays an important role in the recurrence and survival of cancer [16].

Here we present an explorative analysis of muscle and adipose tissue compartments using CT scans to investigate the effects of resistance training on muscle and adipose tissue compartments in the above mentioned randomized controlled SUPPORT-study.

Primary aim was to investigate whether the intervention group showed a better course of body composition with increased muscle tissue compartments than the control group. Secondary aim was to identify predictive factors in body composition that influence the survival of patients with pancreatic cancer.

## Materials & methods

### Study population

Data from the SUPPORT-study (Supervised Progressive Resistance Training for Pancreatic Cancer Patients), a randomized controlled intervention trial investigating the effects of a

6-month lasting progressive resistance training on patients with pancreatic cancer, were used in a post-hoc manner for the present analysis to investigate the effects of the training intervention on muscle and adipose tissue compartments. The study was approved by the Ethics Committee of the Medical Faculty of the University of Heidelberg (S-409/2013) and has been registered on ClinicalTrials.gov (NCT01977066). The methods, the study design and the main results of the SUPPORT-study with regards to the pre-specified primary and secondary outcomes have been published in detail recently [14, 15, 17].

In brief, from 12/2013 until 12/2015 65 out of 304 eligible patients were recruited with following inclusion criteria: age ≥18 years, resectable or non-resectable cancer (stage I-IV), treatment at Heidelberg University Hospital in Germany, sufficient German language skills and informed consent. Patients with adenocarcinoma of the distal bile duct and with ampullary ductal adenocarcinoma were also eligible because of the same medical treatment regime. Following eligibility criteria were changed very early of recruitment to improve the low recruitment rate: patients who performed sports more than 150 minutes per week, stage III and IV and patients who had their surgical resection within the last 12 months were also included. Exclusion criteria were: heart insufficiency more than grade III of the New York Heart Association (NYHA) or uncertain arrhythmia, uncontrolled hypertension, severe renal dysfunction (GFR <30%, creatinine >3 mg/dl), uncompleted wound healing, insufficient hematological capacity (either hemoglobin value <8 g/dl or thrombocytes <50,000), reduced standing or walking ability, or any other comorbidities that precluded their participation.

Patients living close to the study center (<20km) were randomized to a supervised progressive resistance training group (RT1) or to the control group (CON). Patients living further away were randomized to a home-based progressive resistance training group (RT2) or to CON. A 2:1 block randomization, stratified by sex and age, with a random number generator and varying block sizes of 3 and 6 was used. Randomization of a patient was done by an independent biometrician according to the pre-specified allocation list. Assessment for outcome parameters took place prior to the intervention start (T0, baseline) and post-intervention after 6 months (T2). Baseline assessments took place at the earliest 3 months after surgical resection to allow for adequate wound healing. For practicability and safety reasons, parts of the study personnel were unblinded.

## Intervention

RT1 and RT2 performed a resistance training program over a 6-month period with training sessions of approximately 60 minutes twice a week. The sessions included resistance exercises for the major muscle groups of the upper and lower extremities with performance adapted increasing weights. After a four-week adaptation phase, patients performed 8 exercises/session with 2–3 sets with 8–12 repetitions. The training of patients in RT1 took place at an exercise facility at the Heidelberg University's campus on weight machines under supervision of a specialized exercise therapist with exercise intensities of 60–80% One-Repetition Maximum. Patients in RT2 exercised with a training manual on their own at home with exercise intensities of 14–16 on the Borg Scale of Perceived Exertion [18] supported through the exercise therapist by weekly phone calls. Each training session carried out was documented on a training sheet.

CON received usual care in line with their cancer treatment. Patients were called once a month and asked about possible treatment-related side effects and were advised not to change exercise behaviour.

## Outcome assessment

For this analysis CT scans at T0 and T2 were analysed. All of the CT scans were performed in the clinical routine with clinical indication without additional CT scans being performed in

the context of the SUPPORT-study. Inclusion criteria for patients of the SUPPORT-study into this post-hoc analysis were: CT scans suitable in quality and time for T0 and T2 (date of the baseline CT scan -120 days before and +35 days after T0; date for the follow-up CT at T2–35 days before and +35 days after T2), technically evaluable CT scans, level between lumbar vertebral body 3 and 4 (L3/4) included in scans, patient in field of view.

**Quantification of body compartments via CT scans.**   Contrast-enhanced CT scans were retrieved from the institutional PACS (GE Medical Systems, Buckinghamshire, UK) and area-based quantification was performed with a semiautomatic volume tool (Syngo Volume Tool, Siemens Healthineers, Munich, Berlin, Germany). Quantification of body compartments was performed on a single slice between lumbar vertebral body 3 and 4 (L3/4) at the lower endplate of L3 by manually defining specific regions of interest (ROI) [9, 19, 20]. These ROIs were measured using threshold values (in Hounsfield-Units; HU) and the obtained volumes ($cm^3$) were divided by slice-thickness (cm) to get area values ($cm^2$). Of totally 81 CT scans, 90% (n = 73) had a slice-thickness of 0.3cm (5 with 0.5cm; 2 with 0.2cm and 1 with 0.4cm).

The adipose tissue was divided into Total-Fat-Area (TFA), Visceral-Fat-Area (VFA) and Subcutaneous-Fat-Area (SFA). TFA was measured by drawing the ROI around the whole body circumference. VFA was measured by drawing the ROI along the inside of the abdominal wall. The measurement for adipose tissue was restricted to an upper threshold of -30HU and a lower threshold of -190HU [9, 20].

The muscle tissue was quantified on the same slice by drawing a ROI including all muscles on that level (M. erector spinae, M. psoas major, M. rectus abdominis, M. obliquus internus abdominis, M. obliquus externus abdominis, M. transversus abdominis, M. quadratus lumborum, M. latissimus dorsi). The first measurement of Muscle-Area ($MA_{150}$) was performed with a wide range of an upper threshold of +150HU and a lower threshold of -29HU [21, 22], containing the fatty infiltration of muscle tissue as well. The second measurement of Muscle-Area ($MA_{100}$) in the same ROI was with a smaller range of an upper threshold of +100HU and a lower threshold of +40HU, hereby excluding the fatty infiltrated muscle fraction. Mean muscle density of the muscle quantifications in HU was obtained ($MD_{150}$ and $MD_{100}$). Thirdly, the adipose tissue within the muscle-ROI (IMFA, intramuscular-fat-area) was quantified with an upper threshold of -30HU and a lower threshold of -190HU.

SFA was calculated by subtracting VFA and IMFA from the TFA. Visceral-to-subcutaneous-Fat-Ratio (VFR) was calculated by dividing VFA/SFA [9]. Skeletal-Muscle-Index (SMI) was calculated by adjusting $MA_{150}$ with body height ($MA_{150}$/body-height$^2$; Unit $cm^2/m^2$) [21]. Differences of parameters were calculated by: $parameter_{diff} = parameter_{T2} - parameter_{T0}$.

**Strength parameters.**   Muscle strength was assessed bilaterally for extensors and flexors of the elbow, knee and hip with an isokinetic dynamometer (IsoMed2000; D&R Ferstl GmbH, Hemau, Germany). Maximal isokinetic peak torque (MIPT) was assessed with angular velocity of 60˚/s. Patients were instructed to move the machine arm as strong and as fast as they can for 10 repetitions. Maximal voluntary isometric contraction (MVIC) was measured at the strongest angle position each (elbow flexor 80˚, knee extensor 36˚, hip flexor 33˚). Patients were instructed to exert maximum force and to keep it for 6 seconds. Only values of the dominant side were included in the analysis.

Clinical data and patient characteristics were extracted from the medical records or by self-report of the patients. Weight and height were measured during the assessments. Smoking habits and exercise behaviour in the year before the pancreatic cancer diagnosis were assessed by self-report. Patient exercise behaviour was converted into MET/hours per week (metabolic equivalent) according to the Ainsworth compendium of physical activities [23].

**Survival data.**   For survival analysis, data was taken from the hospitals information system I.S-H. med. (SAP, Walldorf, Germany). If available, date of death was retrieved. If date of

death was not available, date of last contact with the hospital was retrieved. Time between date of first diagnosis and death or last contact with the hospital was calculated.

**Statistical analysis.**　Data was collected using Microsoft Office ACCESS and Excel 2010 (Microsoft Corporation, Redmond, WA, USA). Statistical analyses were performed using SPSS Statistics 21 (IBM, Armonk, NY, USA) and SAS Enterprise Guide (version 6.1, SAS statistics, Cary, North Carolina, USA).

For the explorative analysis, RT1 and RT2 were combined to a pooled resistance training group (RT) due to small sample sizes. The dataset included all patients for which evaluable data were available after 6 months (complete-case analysis). Analyses of covariance (ANCOVA) were used to analyse the differences in body composition between groups from pre- to post-intervention. The group assignment (according to intention-to-treat analysis) was used as independent variable, the change since baseline as dependent variable and the baseline measure as covariate. Effect sizes were analysed by computing the partial omega-squared ($\omega^2_p$) coefficient using analyses of covariance. To compare parameters from T0 and T2 paired t-test were used.

For the correlation of CT acquired muscle parameters with muscle strength parameters Spearman correlation coefficients were used. For survival analysis at baseline univariate cox regressions were used. Bivariable Cox regression models were used to assess the risk factor for death using values that have changed over time as time-dependent covariates and intervention group as fixed factor. Kaplan-Meier-curves with log-rank-test were used to compare overall survival of patients with high vs. low VFR and patients with muscle loss vs. muscle gain.

For the presented explorative analysis on routine CT scans no further power calculation was performed.

Results were considered statistically significant at $p<0.05$.

## Results

### Patient characteristics

In total, 53 of 65 randomized pancreatic cancer patients had eligible CT scans at baseline. Out of these, 28 patients completed the 6-month intervention period and showed eligible CT scans at T2, 19 patients in RT and 9 patients in CON (Fig 1). Patient characteristics for all patients (n = 53) as well as for the patients with eligible CT scans before and after the intervention (n = 28) are described in Table 1. Mean age was 62.1 years (SD = 9.0 years) and mean body mass index (BMI) was 23.9 kg/m$^2$ (SD = 4.1 kg/m$^2$). Overall the most common cancer type was pancreatic ductal adenocarcinoma (88.7%) and most patients were diagnosed with stage II (77.4%). The most common treatment regime was surgery combined with adjuvant chemo-therapy (83.0%). Most patients were non-smoker (83.0%). The training adherence rate dropped steadily over the 6 months from initially 81.7% to 62.9%. On average, patients performed 1.4 weekly training sessions out of 2. One adverse event occurred, incisional hernia temporally after baseline assessment (CON), no adverse events occurred during exercise sessions.

### Change in body composition

**Adipose tissue compartments.**　Table 2 presents the distribution and change in body composition of adipose tissue compartments of RT and CON from T0 to T2.

There were no between-group differences for the assessed adipose tissue parameters at 6 months (p>0.05; ($\omega^2_p$<0.02). Descriptively, CON showed higher values for CT quantified parameters TFA, VFA and SFA as well as for body weight and BMI at baseline. VFR was slightly higher in RT. During the intervention, RT and CON showed a similar decrease in

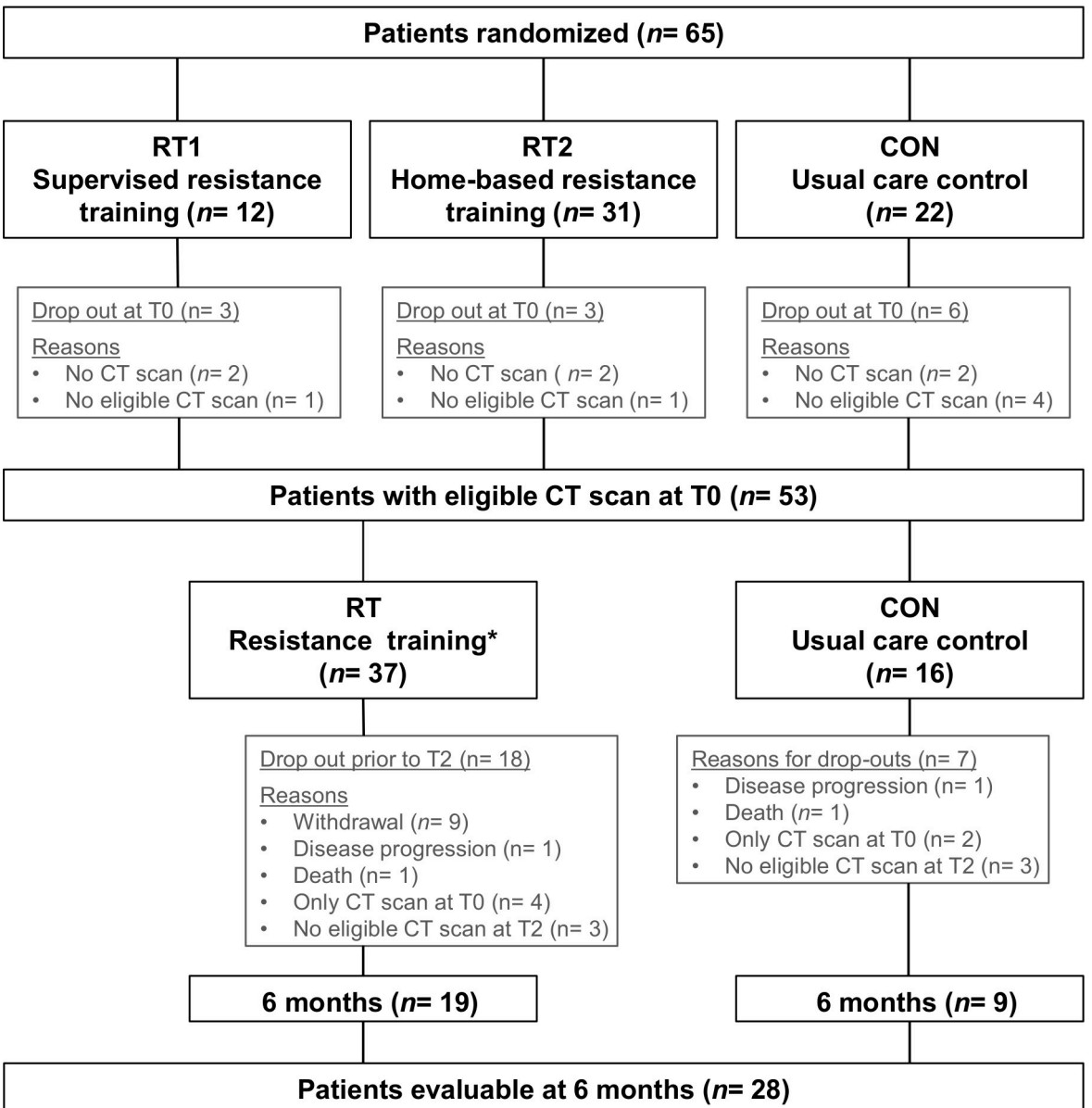

**Fig 1. Patient flow chart.** T0 = baseline; T2 = after 6-month resistance training; * Combining RT1 and RT2.

TFA, VFA and SFA. For VFR, no change was observed in RT, while a slight decrease was observed in CON (-0.1; Table 2). RT showed a slight increase of BMI and body weight while CON showed a slight decrease of BMI and body weight.

**Muscle tissue compartments.** Table 3 presents the distribution and change in body composition of muscle tissue compartments of RT and CON from T0 to T2.

All assessed muscle tissue parameters showed no between-group differences at 6 months ($p > 0.05$; ($\omega^2_p < 0.02$). Descriptively, CON showed higher values for all muscle parameters but $MA_{100}$ and $SMI_{100}$ compared to RT at baseline. From baseline to the end of the 6-month intervention period RT and CON showed a noticeable increase of $MA_{100}$ and $SMI_{100}$, while there was just a small increase of muscle density ($MD_{150}$, $MD_{100}$) in both groups. For $MA_{150}$ and $SMI_{150}$ CON showed a higher increase than RT. IMFA decreased in both groups equally.

**Table 1. Patient characteristics of all patients and divided by group.**

| | All patients | | RT | | CON | |
|---|---|---|---|---|---|---|
| | (n = 53) | | (n = 28) | | | |
| **TOTAL**, n (%) | 53 | (100) | 19 | (100) | 9 | (100) |
| **Age**, years, mean (SD) | 62.1 | (9.0) | 61.0 | (9.2) | 60.6 | (7.9) |
| **Gender**, n (%) | | | | | | |
| Male | 33 | (62.3) | 13 | (68.4) | 6 | (66,7) |
| Female | 20 | (37.7) | 6 | (31.6) | 3 | (33.3) |
| **BMI**, mean (SD) | 23.9 | (4.1) | 23.3 | (3.3) | 25.7 | (2.9) |
| **Cancer Type**, n (%) | | | | | | |
| Pancreatic ductal adenocarcinoma | 47 | (88.7) | 17 | (89.5) | 7 | (77.8) |
| Distal bile duct adenocarcinoma | 4 | (7.5) | 2 | (10.5) | 1 | (11.1) |
| Papillary ductal adenocarcinoma | 2 | (3.8) | | | 1 | (11.1) |
| **Tumor stage**, n (%) | | | | | | |
| Not available | 3 | (5.7) | 1 | (5.3) | | |
| IA | 1 | (1.9) | | | | |
| IB | 6 | (11.3) | 2 | (10.5) | 2 | (22.2) |
| IIA | 7 | (13.2) | 4 | (21.1) | 2 | (22.2) |
| IIB | 34 | (64.2) | 11 | (57.9) | 5 | (55.6) |
| IV | 2 | (3.8) | 1 | (5.3) | | |
| **Operative procedures**, n (%) | | | | | | |
| Total pancreatectomy | 6 | (11.3) | 3 | (15.8) | | |
| Distal pancreatectomy | 8 | (15.1) | 2 | (10.5) | 1 | (11.1) |
| Whipple | 16 | (30.2) | 5 | (26.3) | 3 | (33.3) |
| Pylorus-preserving Whipple | 20 | (37.7) | 8 | (42.1) | 5 | (55.6) |
| No operation | 3 | (5.7) | 1 | (5.3) | | |
| **Treatment**, n (%) | | | | | | |
| Surgery, adj. CHT | 44 | (83.0) | 15 | (78.9) | 9 | (100) |
| Neoadj. CHT, Surgery | 2 | (3.8) | 2 | (10.5) | | |
| Neoadj. CHT, Surgery, adj. CHT | 3 | (5.7) | 1 | (5.3) | | |
| CHT | 3 | (5.7) | 1 | (5.3) | | |
| Surgery | 1 | (1.9) | | | | |
| **Smoking**, n (%) | | | | | | |
| Non-smoker | 40 | (75.5) | 14 | (73.7) | 9 | (100) |
| Recent smoker | 10 | (18.9) | 3 | (15.8) | | |
| Still smoker | 3 | (5.7) | 2 | (10.5) | | |
| **Exercise in the year before diagnosis**, n (%) | | | | | | |
| None | 24 | (45.3) | 9 | (47.4) | 6 | (66.7) |
| 0 - <9 MET*h/week | 10 | (18.9) | 3 | (15.8) | 1 | (11.1) |
| 9 - <18 MET*h/week | 10 | (18.9) | 5 | (26.3) | 2 | (22.2) |
| ≥ 18 MET*h/week | 7 | (13.2) | 2 | (10.5) | | |
| Missing | 2 | (3.8) | | | | |
| **Time between CT scans**, months, mean (SD) | - | - | 7.2 | (1.7) | 7.4 | (1.8) |
| **Surgery between CT scans**, n (%) | - | - | | | | |
| In between | | | 3 | (15.8) | 3 | (33.3) |
| Before | | | 16 | (84.2) | 6 | (66.6) |

Baseline patient characteristics of patients with CT at T0 (n = 53) and patients with CT at T0 and T2 (n = 28) classified by progressive resistance training group (RT) and control group (CON). MET = metabolic equivalent; CHT = chemotherapy; SD = standard deviation; CT = computed tomography.

**Table 2. Distribution of adipose tissue across compartments before (T0) and after 6-month resistance training (T2).**

| Outcome | Group | N | T0 Mean (SD) | T2 Mean (SD) | Adjusted mean change* (95% CI) from T0 to T2 | Adjusted difference between groups, mean (95% CI) | | p‡ | $\omega^2_p$‡‡ |
|---|---|---|---|---|---|---|---|---|---|
| TFA (cm$^2$) | RT | 19 | 299.0 (136.1) | 242.6 (120.6) | -75.8 (-127, -24.5) | RT-CON | -47.3 (-143, 48.0) | 0.317 | 0.002 |
| | CON | 9 | 411.0 (123.9) | 341.4 (120.0) | -28.5 (-105, 48.2) | | | | |
| VFA (cm$^2$) | RT ° | 19 | 134.8 (82.6) | 96.9 (50.8) | -45.4 (-68.4, -22.4) | RT-CON | -22.3 (-63.6, 19.0) | 0.276 | 0.009 |
| | CON ° | 9 | 174.0 (74.1) | 135.2 (70.2) | -23.1 (-56.8, 10.6) | | | | |
| SFA (cm$^2$) | RT | 19 | 153.4 (65.0) | 137.1 (74.1) | -24.3 (-52.9, 4.3) | RT-CON | -12.6 (-67.1, 41.8) | 0.636 | -0.028 |
| | CON | 9 | 220.6 (61.0) | 192.0 (59.1) | -11.7 (-55.0, 31.7) | | | | |
| VFR | RT | 19 | 0.9 (0.4) | 0.9 (0.5) | -0.0 (-0.2, 0.2) | RT-CON | 0.1 (-0.2, 0.5) | 0.441 | -0.014 |
| | CON ° | 9 | 0.8 (0.2) | 0.7 (0.3) | -0.1 (-0.4, 0.1) | | | | |
| BMI (kg/m$^2$) | RT | 17 | 23.5 (3.2) | 23.8 (3.6) | 0.3 (-0.5, 1.1) | RT-CON | 0.6 (-0.8, 2.0) | 0.361 | -0.005 |
| | CON | 8 | 25.5 (3.0) | 25.2 (3.5) | -0.3 (-1.5, 0.8) | | | | |
| Body weight (kg) | RT | 17 | 72.8 (9.2) | 73.7 (11.0) | 1.1 (-1.1, 3.3) | RT-CON | 1.9 (-2.0, 5.8) | 0.330 | -0.000 |
| | CON | 8 | 78.3 (13.9) | 77.9 (15.9) | -0.8 (-4.0, 2.4) | | | | |

ANCOVA; n = 28; compartments quantified at level L3/4. TFA = total fat area, VFA = visceral fat area, SFA = subcutaneous fat area, VFR = visceral fat ratio, BMI = body mass index, RT = resistance training group, CON = usual care control group

* Adjusted for baseline value

‡ diff

‡‡ effect size partial omega squared

° Significant differences T0 vs. T2 (paired t-test; p<0.05).

**Table 3. Distribution of muscle tissue compartments and mean attenuation before (T0) and after 6-month resistance training (T2).**

| Outcome | Group | N | T0 Mean (SD) | T2 Mean (SD) | Adjusted mean change* (95% CI) from T0 to T2 | Adjusted difference between groups, mean (95% CI) | | p‡ | $\omega^2_p$‡‡ |
|---|---|---|---|---|---|---|---|---|---|
| MA$_{150}$ (cm$^2$) | RT | 19 | 143.5 (26.1) | 143.7 (28.8) | 0.3 (-5.7, 6.2) | RT-CON | -5.4 (-16.0, 5.1) | 0.298 | 0.005 |
| | CON | 9 | 146.2 (32.3) | 151.8 (33.3) | 5.7 (-3.0, 14.4) | | | | |
| MA$_{100}$ (cm$^2$) | RT | 19 | 97.5 (21.0) | 106.4 (30.0) | 9.4 (-0.8, 19.6) | RT-CON | -1.7 (-20.3, 16.9) | 0.851 | -0.036 |
| | CON | 9 | 82.9 (21.8) | 95.1 (22.9) | 11.1 (-3.9, 26.2) | | | | |
| IMFA (cm$^2$) | RT ° | 19 | 10.7 (5.0) | 8.6 (4.4) | -3.1 (-4.9, -1.3) | RT-CON | -2.9 (-6.4, 0.6) | 0.097 | 0.066 |
| | CON | 9 | 16.4 (5.5) | 14.2 (4.1) | -0.2 (-2.9, 2.6) | | | | |
| SMI$_{150}$ (cm$^2$/m$^2$) | RT | 19 | 46.3 (7.3) | 46.4 (8.5) | 0.1 (-1.8, 2.0) | RT-CON | -1.8 (-5.2, 1.6) | 0.288 | 0.006 |
| | CON | 9 | 47.5 (8.3) | 49.4 (8.5) | 1.9 (-0.9, 4.7) | | | | |
| SMI$_{100}$ (cm$^2$/m$^2$) | RT | 19 | 31.4 (6.0) | 34.3 (8.9) | 3.1 (-0.1, 6.4) | RT-CON | -0.3 (-6.3, 5.7) | 0.909 | -0.037 |
| | CON | 9 | 27.0 (6.0) | 31.0 (6.5) | 3.5 (-1.4, 8.3) | | | | |
| MD$_{150}$ (HU) | RT | 19 | 46.4 (7.1) | 48.9 (6.5) | 3.7 (1.1, 6.4) | RT-CON | 3.0 (-2.1, 8.1) | 0.233 | 0.017 |
| | CON | 9 | 39.6 (3.2) | 42.9 (4.3) | 0.7 (-3.3, 4.7) | | | | |
| MD$_{100}$ (HU) | RT | 19 | 58.7 (3.5) | 59.5 (2.8) | 1.4 (0.3, 2.6) | RT-CON | 1.0 (-1.1, 3.2) | 0.341 | -0.002 |
| | CON | 9 | 56.2 (1.6) | 57.8 (1.8) | 0.4 (-1.3, 2.1) | | | | |

ANCOVA; n = 28; compartments quantified at level L3/4. MA = muscle area, IMFA = inter-muscular-fat area, SMI = skeletal muscle index, MD = muscle density (in HU), RT = resistance training group, CON = usual care control group

* Adjusted for baseline value

‡ diff

‡‡ effect size partial omega squared; ° Significant differences T0 vs. T2 (paired t-test; p<0.05).

## Surgery between CT scans

22 out of 28 patients (78.6%) had surgery before the baseline CT and showed a significant increase in muscle parameters between T0 and T2 (see S1 Table; difference in SMI = 1.4; p = 0.03). No significant change in fat parameters was observed. 6 patients (21.4%) had surgery after the baseline CT (see S2 Table) and therefore between the CT scans. Those patients showed a significant decrease in fat parameters (difference of TFA = -219.8; p = 0.013). No significant difference in muscle parameters was found.

## Muscle mass and muscle strength

Table 4 presents the correlations of the measured muscle strength parameters and the CT acquired muscle parameters using Spearman correlation coefficients. The calculation was performed with baseline values at T0, n = 53.

There were moderate to high positive correlations between $MA_{150}$ and the muscle strength parameters (r = 0.57–0.85; p<0.001). Between $SMI_{150}$ and muscle strength parameters significant low to moderate positive correlations (r = 0.39–0.68, p<0.01) were observed.

$MA_{100}$ and $SMI_{100}$ showed a moderate to high positive correlation with the muscle strength parameters ($MA_{100}$ r = 0.51–0.72; p<0.001; $SMI_{100}$ r = 0.41–0.55; p<0.01).

For muscle density ($MD_{150}$ and $MD_{100}$) and intramuscular fat (IMFA), no correlation was observed with the muscle strength parameters (r = -0.2–0.23; p>0.05).

**Table 4. Correlations of muscle strength parameters with CT acquired muscle parameters.**

|  | Knee extensors | Knee extensors | Elbow flexors | Elbow flexors | Hip flexors | Hip flexors |
|---|---|---|---|---|---|---|
|  | MIPT | MVIC | MIPT | MVIC | MIPT | MVIC |
| **$MA_{150}$ (cm$^2$)** | **0.71** | **0.73** | **0.82** | **0.85** | **0.62** | **0.57** |
| p-value | < .001* | < .001* | < .001* | < .001* | < .001* | < .001* |
| n | 53 | 53 | 53 | 53 | 51 | 51 |
| **$SMI_{150}$ (cm$^2$/m$^2$)** | **0.50** | **0.54** | **0.66** | **0.69** | **0.45** | **0.40** |
| p-value | 0.001* | < .001* | < .001* | < .001* | <0.001** | 0.004* |
| n | 53 | 53 | 53 | 53 | 51 | 51 |
| **$MD_{150}$ (HU)** | **0.23** | **0.15** | **0.17** | **0.18** | **0.20** | **0.21** |
| p-value | 0.099 | 0.292 | 0.211 | 0.191 | 0.158 | 0.136 |
| n | 53 | 53 | 53 | 53 | 51 | 51 |
| **$MA_{100}$ (cm$^2$)** | **0.65** | **0.63** | **0.70** | **0.73** | **0.57** | **0.52** |
| p-value | < .001* | < .001* | < .001* | < .001* | < .001* | <0.001* |
| n | 53 | 53 | 53 | 53 | 51 | 51 |
| **$SMI_{100}$ (cm$^2$/m$^2$)** | **0.52** | **0.48** | **0.55** | **0.56** | **0.47** | **0.42** |
| p-value | < .001* | <0.001* | < .001* | < .001* | <0.001* | 0.002* |
| n | 53 | 53 | 53 | 53 | 51 | 51 |
| **$MD_{100}$ (HU)** | **-0.01** | **-0.12** | **-0.12** | **-0.10** | **-0.06** | **-0.03** |
| p-value | 0.964 | 0.411 | 0.405 | 0.458 | 0.671 | 0.855 |
| n | 53 | 53 | 53 | 53 | 51 | 51 |
| **IMFA (cm$^2$)** | **0.04** | **0.05** | **0.01** | **0.02** | **-0.13** | **-0.21** |
| p-value | 0.770 | 0.716 | 0.967 | 0.886 | 0.365 | 0.145 |
| n | 53 | 53 | 53 | 53 | 51 | 51 |

All patients at baseline (T0), n = 53. Calculation of Spearman correlation coefficients. MIPT: maximal isokinetic peak torque (in Newton Meter); MVIC: maximal voluntary isometric contraction (in Newton); MA = muscle area, SMI = skeletal muscle index, MD = muscle density (in HU), IMFA = inter-muscular-fat area

* = significant

° = strength-measurement of hip flexors could not be performed in n = 2 patients, thus correlation of hip flexion was calculated with n = 51 patient.

**Table 5. Univariate survival analysis with baseline parameters.**

| Parameter T0 | HR | 95% CI lower | upper | p-value |
|---|---|---|---|---|
| BMI | 0.958 | 0.857 | 1.071 | 0.451 |
| TFA | 0.999 | 0.996 | 1.002 | 0.572 |
| VFA | 1.000 | 0.995 | 1.005 | 0.933 |
| SFA | 0.997 | 0.992 | 1.002 | 0.288 |
| IMFA | 0.978 | 0.907 | 1.055 | 0.563 |
| VFR | 2.084 | 1.163 | 3.732 | 0.014* |
| $MA_{150}$ | 0.992 | 0.978 | 1.006 | 0.273 |
| $MD_{150}$ | 1.019 | 0.962 | 1.078 | 0.528 |
| $SMI_{150}$ | 0.982 | 0.933 | 1.034 | 0.492 |
| $MA_{100}$ | 0.996 | 0.980 | 1.013 | 0.662 |
| $MD_{100}$ | 1.095 | 0.964 | 1.244 | 0.162 |
| $SMI_{100}$ | 0.996 | 0.942 | 1.052 | 0.883 |

Cox regressions and calculations hazard-ratios (HR), n = 53. CI: confidence interval; * = significant.

## Survival analysis

The survival analysis for the baseline values was performed with all n = 53 patients included. Table 5 presents the univariate cox regressions with the parameters at T0.

At baseline, VFR showed a significant influence on the overall survival (HR = 2.084; p = 0.014). Hereby a high VFR indicated a higher risk of death. The other adipose and muscle tissue parameters showed no significant influence on overall survival. Fig 2 shows the Kaplan-Meier-curve comparing patients with high VFR ($\geq$1.3; n = 8) showing a lower median overall survival of 14.6 months with patients with lower VFR (<1.3; n = 45) having a higher median overall survival of 45.3 months (p = 0.012).

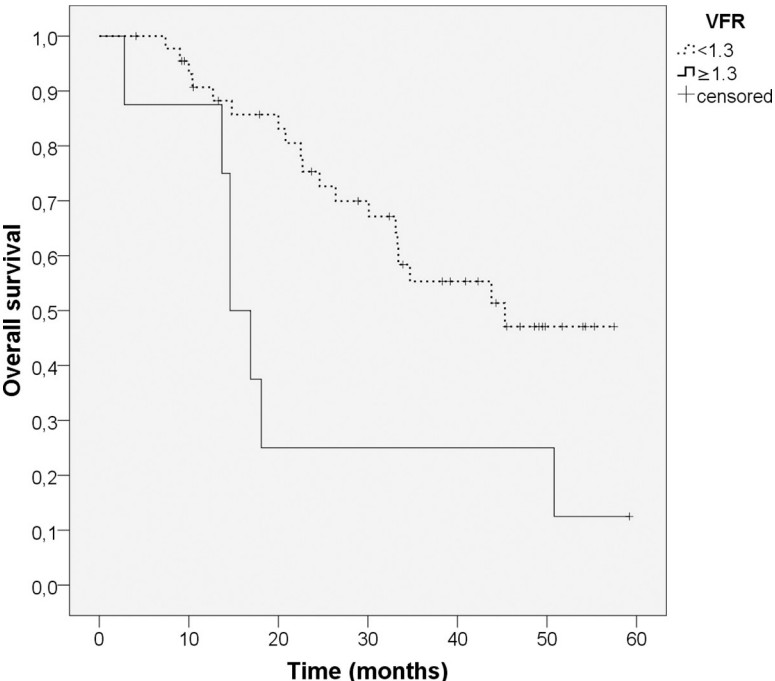

**Fig 2. Kaplan-Meier-curve for VFR at T0.** Log-rank-test, n = 53. Patients with high VFR ($\geq$1.3; n = 8; continuous line) show a lower median overall survival of 14.6 months than patients with low VFR (<1.3; n = 45; dotted line) with a median overall survival of 45.3 months (p = 0.012).

**Table 6. Bivariate survival analysis with difference of parameters from T0 to T2.**

| Parameter T2-T0 | HR | 95% CI lower | upper | p-value |
|---|---|---|---|---|
| BMI (n = 24) | 0.815 | 0.671 | 0.990 | 0.040 * |
| TFA | 0.998 | 0.994 | 1.003 | 0.432 |
| VFA | 0.998 | 0.988 | 1.007 | 0.624 |
| SFA | 0.996 | 0.989 | 1.004 | 0.328 |
| IMFA | 1.035 | 0.905 | 1.182 | 0.618 |
| VFR | 1.041 | 0.324 | 3.340 | 0.947 |
| $MA_{150}$ | 0.986 | 0.967 | 1.006 | 0.174 |
| $MD_{150}$ | 0.959 | 0.875 | 1.051 | 0.371 |
| $SMI_{150}$ | 0.940 | 0.878 | 1.018 | 0.143 |
| $MA_{100}$ | 0.983 | 0.962 | 1.005 | 0.137 |
| $MD_{100}$ | 1.000 | 0.821 | 1.219 | 0.997 |
| $SMI_{100}$ | 0.943 | 0.876 | 1.014 | 0.113 |

Cox regressions (adjusted for time as dependent variable and intervention group as fixed factor) and calculation of hazard-ratios (HR), n = 28. CI: confidence interval

* = significant.

The survival analysis for the change of parameters between T0 and T2 was performed with n = 28 patients. Table 6 presents the time-dependent Cox regressions with the differences of parameters between T0 and T2. There was no significant influence of the changes of the adipose and muscle tissue parameters on overall survival. For the change of BMI from T0 to T2 a significant influence on the overall survival was observed. Fig 3 shows the Kaplan-Meier-curve comparing patients with loss of muscle mass ($SMI_{150}$-difference $< 0 cm^2/m^2$) with patients that

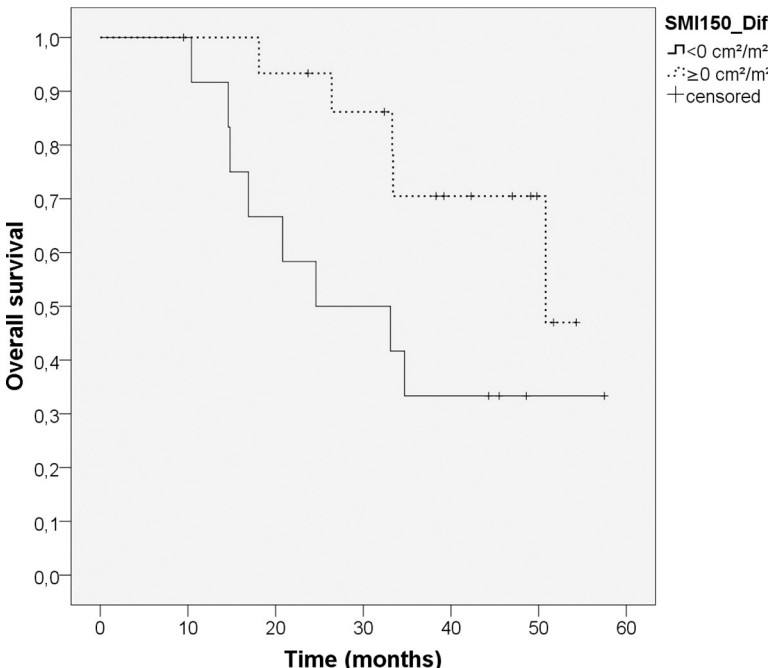

**Fig 3. Kaplan-Meier-curve for difference in $SMI_{150}$ from T0 to T2.** Log-rank-test, n = 28. Patients with loss of muscle mass ($SMI_{150}$-difference $< 0 cm^2/m^2$; n = 12, continuous line) show a median overall survival of 24.6 months vs. patients with gain of muscle mass ($SMI_{150}$-difference $\geq 0 cm^2/m^2$, n = 16; dotted line) and a median survival of 50.8 months (p = 0.049).

gained muscle mass ($SMI_{150}$-difference $\geq 0$ $cm^2/m^2$). Patients with muscle loss showed a lower overall survival than patients with muscle gain (24.6 months vs. 50.8 months, p = 0.049).

## Discussion

The explorative analysis presented here investigated the effects of progressive resistance training on muscle and adipose tissue compartments in pancreatic cancer patients. Our primary aim was to investigate, if RT shows a better course of body composition with a higher increase in muscle mass. After the 6-month intervention period, there were no significant differences of resistance training on muscle and adipose tissue parameters between RT and CON. At the same time, the data show no depletion of muscle mass in all patients. On the contrary, there was a very slight increase in muscle mass, although it was greater in CON than in RT. Therefore, we couldn't confirm our primary hypothesis in this study. In addition, we observed a good correlation between muscle strength and muscle mass. Our secondary aim was to identify prognostic parameters for overall survival. Both a high visceral to subcutaneous fat ratio and loss of muscle mass turned out to be predictors of poor overall survival.

It is frequently reported that patients with pancreatic cancer show sarcopenia and a decrease of muscle mass, which has been associated with negative effects and a poor prognosis [24–26]. The question emerged if it is possible to counter these catabolic effects with an exercise training program. In our study population we could show, that there is an anabolic potential in patients with pancreatic cancer. Patients in RT maintained their muscle mass with a very slight increase during the intervention period ($MA_{150}$ +0.3cm; $SMI_{150}$ +0.1 $cm^2/m^2$; see Table 3). Therefore, resistance training might have a positive impact on maintaining muscle mass in pancreatic cancer patients. However, patients in CON on average even gained fair amounts of muscle mass ($MA_{150}$ +5.7 cm; $SMI_{150}$ +1.9 $cm^2/m^2$; see Table 3) although they were not part of a training program. We cannot really explain the fact that CON showed a higher increase in muscle mass than RT and our explanations for this topic are speculative. One possible explanation could be that hormones, that play a role in anabolism and catabolism, such as insulin produced in the pancreas, are disrupted through surgery and training effects and thus have an altered effect on the overall metabolism. The patients with pancreatic cancer underwent pancreatic surgery in varying extent with changes and disruption of production of insulin in the pancreas. Patients adapt differently and individually to these changes. Insulin is an anabolic hormone and plays a big role in protein synthesis and muscle metabolism [27]. Therefore the changes in insulin production may be an influencing factor on the change in muscle mass. Another possible, albeit very speculative explanation could be a contamination of the control group, e.g. the adoption of the intervention by themselves [28, 29]. It can be assumed that most patients taking part in an exercise intervention study are highly motivated to participate in the intervention and have generally a positive attitude towards exercise. Patients of CON could catch up information of the exercise program through other patients, medical staff or literature/media. Consequently, they could become motivated to do some exercises themselves. However, CON would have had to do the same training load as the RT or even more in order to achieve a significant effect compared to the RT, which seems unlikely.

There was a decrease of adipose tissue across all quantified compartments in control and resistance training group during the intervention period. One reason for this loss of adipose tissue could be cachexia with catabolic processes due to the malignant disease. Further, treatment of the patients with chemotherapy and surgery might have negative effects on body composition, which might be another probable explanation for the loss of adipose tissue. In addition, both groups showed no significant change in BMI over this time. This indicates that

some changes in body composition are not detected by the anthropometric measurements like weight or BMI alone. Those changes may occur before they are detected by those measurement tools and thus this possible sign of cachexia might be registered earlier by imaging [30].

To our knowledge, there is currently no directly comparable exercise intervention study available with patients with pancreatic cancer and CT quantified body compartments. Dieli-Conwright et al. analysed the effect of a 16-week combined aerobic and resistance exercise program on breast cancer survivors [31]. They found a decrease in body weight and an increase in lean body mass and appendicular skeletal muscle index [31]. Another study investigated the effects of a 12-week resistance training on body composition in prostate cancer patients [32]. They also found increases in lean body mass and reduced sarcopenia [32]. In both studies muscle mass was quantified by dual-energy X-ray absorptiometry (DXA). Despite the different method and different cancer types this agrees with our finding, that there is anabolic potential in cancer patients and that an exercise intervention program may increase muscle mass. These findings may be less distinctive in patients with pancreatic cancer, because of the higher malignant potential of pancreatic cancer and a larger surgical procedure.

Our results showed a strong correlation between muscle strength and muscle mass (Table 4). Patients with more muscle mass tended to show a higher strength. Especially the $MA_{150}$ including also the fatty infiltrated muscle parts revealed a strong correlation with the elbow flexors (r = 0.85; Table 4) and the knee extensors (r = 0.73). $MA_{100}$ also showed a good but lower correlation with the muscle strength parameters than $MA_{150}$. $MA_{100}$ was quantified with tighter threshold values (in HU), which excluded the fatty infiltrated muscle parts, which had a lower HU attenuation because of the partial volume effect. Therefore, the fatty infiltrated muscle parts seem to contribute to the muscle strength which could explain the higher correlation coefficients. Also for the skeletal muscle index ($SMI_{150}$, $SMI_{100}$) and muscle strength a good positive correlation was observed. The muscle density parameters ($MD_{150}$ and $MD_{100}$) and the IMFA showed no significant correlation with muscle strength. This indicates that muscle density as well as fat tissue between the various muscle parts (IMFA) are not the primary factors for muscle strength. Our results are partially consistent with a study by MacDonald et al. [33], which measured the correlation between CT quantified L3 SMI and lower limb muscle strength and measures of complex function. Although they did not see a correlation between lower limb muscle strength and L3 SMI but only between complex functions and SMI, they also found a correlation between muscle mass and lower limb muscle strength (MRI quantified mass of M. quadriceps femoris). Although these findings don't match completely with our results, they point in a similar direction. $SMI_{150}$ at lumbar L3/4 level therefore may represent a surrogate for general muscle strength and function.

At baseline, patients with a high VFR tended to have a poorer survival than patients with a lower VFR (median survival 14.6 vs. 45.3 months; p = 0.012; Fig 2). A high VFR represents a high amount of intraabdominal fat tissue (VFA) in relation to the subcutaneous fat tissue (SFA). This finding of a prognostic impact of high VFR aligns with other studies conducted with pancreatic cancer patients [34, 35] or lung cancer patients [9], where a high VFR turned out to be a predictor of poor prognosis. This prognostic impact seems to be consistent across several cancer entities. Visceral fat tissue seems to have a different risk profile than subcutaneous fat tissue [36]. Additionally, it was previously reported that patients with a high amount of visceral fat tissue also have a higher cardiovascular mortality [36].

Muscle mass ($MA_{150}$) at baseline didn't show a statistically significant impact on overall survival in our study population. Low muscle mass has previously been reported to be predictive of poor overall survival in resectable and advanced pancreatic cancer [24–26, 37, 38]. Several other studies couldn't show a predictive impact of muscle mass at a single time point with pancreatic cancer and lung cancer [9, 39]. Nevertheless, a high absolute amount of muscle

mass still seems advantageous. Muscle density ($MD_{150}$) at baseline also didn't turn out to be a predictor of overall survival in our study population. Lower mean muscle density (in HU-values) of muscle tissue is presumably caused by fatty infiltration of muscle tissue and may be a sign of muscle wasting. Some studies showed an impact of muscle density on overall survival [21, 40], while others showed no impact [9]. It may be due to small sample size in our study population that there was no statistically significant effect on prognosis. Patients with a loss of muscle mass over time (mean 7.3 months) showed a poorer overall survival than patients without muscle loss (median survival 24.6 vs. 50.8 months; p = 0.049; see Fig 3). Loss of muscle mass is a central element of the cachectic syndrome [5] and was previously reported to predict poorer survival in patients with pancreatic cancer [24, 41] as well as with lung cancer [9].

This was a explorative sub-analysis of CT quantified muscle and adipose tissue compartments within the randomized controlled SUPPORT-study. Patients performed progressive resistance training, muscle strength parameters as well as muscle and adipose tissue compartments were assessed using criterion-standard assessments and intention-to-treat analysis was performed. However, our study had also some limitations. First, the small sample size of 53 patients with eligible CT scans in total, respectively, 28 patients with eligible CT scans who completed the 6-month intervention period, resulting in a reduced generalizability. The SUPPORT-study was stopped due to recruitment difficulties, thus, before the planned sample size of 150 evaluable patients had been reached. The drop-out rate was as expected and similar in the 3 groups. Further, the unbalanced group size of RT and CON could be an influencing factor. Due to the allocation of patients to either of the two resistance training groups, supervised or home-based, or to CON according to the living distance of the patients and the fact that more distant living patients were included, unbalanced group sizes occurred. In addition, for the presented analyses both resistance training groups were combined to one resistance training group due to the small number of patients, so the results should be interpreted carefully. Another limitation was that muscle strength was measured by muscle groups of the upper and lower extremities and muscle tissue parameters at the lumbar level L3/4 of the body trunk. Thus, the respective muscle parameters recorded did not match. In addition, a small number of patients (21.4%) had surgery performed in between CT scans as a possible influencing factor on muscle and adipose tissue compartments (see S1 and S2 Tables).

In conclusion, we couldn't confirm our primary hypothesis in this study that RT shows a greater increase in muscle mass than CON, due to an increase in muscle mass in CON. But the results of our study support the assumption that there is an anabolic potential in patients with pancreatic cancer. RT sustained muscle mass and CON gained a small amount of muscle mass. Progressive resistance training may be a promising modality to support pancreatic cancer patients to maintain their muscle mass and avoid muscle depletion. The parameters used in this study could also help identify patients, who may profit from progressive resistance training, and monitor the progress during the training period. Furthermore, muscle mass quantified at L3/4 showed a good correlation to muscle strength. Therefore, maintaining muscle mass and muscle strength through structured resistance training could help patients to maintain their physical functioning. For our secondary aim, to identify prognostic parameters, we found that a high loss of muscle mass and a high visceral to subcutaneous fat ratio at baseline turned out to be predictors of poor overall survival. Therefore, the CT-quantified parameters could help with risk stratification of patients. Corresponding training programs could help to avoid or weaken these signs of cachexia in pancreatic cancer patients. Further randomized controlled exercise intervention studies with higher numbers of patients and bigger control groups should be conducted to verify possible benefits of maintaining muscle mass. Additional randomized and controlled studies are also needed to determine the optimal

intensity and quantity of training programs to achieve possible positive effects for patients with pancreatic cancer.

## Supporting information

**S1 Checklist.**
(DOC)

**S1 Table. CT quantified body compartments with a baseline CT after surgery.** n = 22. TFA = total fat area, VFA = visceral fat area, SFA = subcutaneous fat area, VFR = visceral fat ratio, MA = muscle area, IMFA = inter-muscular-fat area, SMI = skeletal muscle index, MD = muscle density (in HU); paired t-test; * = significant.
(DOCX)

**S2 Table. CT quantified body compartments with a Baseline CT before surgery.** n = 6. TFA = total fat area, VFA = visceral fat area, SFA = subcutaneous fat area, VFR = visceral fat ratio, MA = muscle area, IMFA = inter-muscular-fat area, SMI = skeletal muscle index, MD = muscle density (in HU); paired t-test; * = significant.
(DOCX)

**S1 File. Study protocol SUPPORT-study.**
(PDF)

## Acknowledgments

The authors thank the patients who participated in this clinical trial.

## Author Contributions

**Conceptualization:** Raoul Wochner, Dorothea Clauss, Johanna Nattenmüller, Thilo Hackert, Joachim Wiskemann, Karen Steindorf.

**Data curation:** Raoul Wochner, Dorothea Clauss.

**Formal analysis:** Raoul Wochner, Dorothea Clauss, Thomas Bruckner.

**Funding acquisition:** Joachim Wiskemann, Karen Steindorf.

**Investigation:** Raoul Wochner, Dorothea Clauss, Johanna Nattenmüller, Joachim Wiskemann, Karen Steindorf.

**Methodology:** Raoul Wochner, Dorothea Clauss, Johanna Nattenmüller, Thomas Bruckner, Thilo Hackert, Karen Steindorf.

**Project administration:** Dorothea Clauss, Johanna Nattenmüller, Karen Steindorf.

**Resources:** Christine Tjaden, Hans-Ulrich Kauczor, Joachim Wiskemann, Karen Steindorf.

**Software:** Raoul Wochner, Dorothea Clauss, Thomas Bruckner.

**Supervision:** Johanna Nattenmüller, Hans-Ulrich Kauczor, Thilo Hackert, Joachim Wiskemann, Karen Steindorf.

**Validation:** Raoul Wochner, Dorothea Clauss.

**Visualization:** Raoul Wochner, Dorothea Clauss.

**Writing – original draft:** Raoul Wochner, Dorothea Clauss, Johanna Nattenmüller.

**Writing – review & editing:** Raoul Wochner, Dorothea Clauss, Johanna Nattenmüller, Christine Tjaden, Thomas Bruckner, Hans-Ulrich Kauczor, Thilo Hackert, Joachim Wiskemann, Karen Steindorf.

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
