## [Decision Letter · Decision Letter 0]

1 Jul 2020

PONE-D-20-06946

Impact of progressive resistance training on CT quantified muscle and adipose tissue compartments in pancreatic cancer patients

PLOS ONE

Dear Dr. Steindorf,

Thank you for submitting your manuscript to PLOS ONE. After careful consideration, we feel that it has merit but does not fully meet PLOS ONE’s publication criteria as it currently stands. Therefore, we invite you to submit a revised version of the manuscript that addresses the points raised during the review process.

While I find the study well-written and organized, some major concerns were raised by the reviewers and by me. Please, consider replying carefully all questions.

Please, send the dataset to specific supplementary material server managed by the library service of the German Cancer Research Center.

Abstract, line 29: which hypothesis was tested? (please state it).

Abstract, lines 29-31: Please put the primary outcome first to indicate the hierarchy. Please also clearly state what was the primary outcome.

Abstract, results: Please add effect sizes and not just p-values. Focus on group contrasts in the reporting (please see the PREPARE Trial guide for guidance). And, consider including ES and p-values in the non-differences found.

Abstract, end: Please add clinical-trial registration-info at the end of the abstract. Because it seems as if the trial was retrospectively registered (registration after inclusion of the first participant) add “retrospectively registered” after the trial registration number. Please state clearly in the manuscript if the primary outcome was pre-defined (defined before inclusion of the first participant).

Hypotheses

Consider including objective hypotheses, not just diff or not, but y higher than x condition style.

Was it possible to account for confounding factors across groups, such as the effect of pharmacological doses? 

Please remove statistical tests for baseline differences. CONSORT advise against this. Please see http://www.consort-statement.org/Media/Default/Downloads/CONSORT%202010%20Explanation%20and%20Elaboration%20Document-BMJ.pdf  page 17.

Results and Stats: please report 95CI of all variables and effect sizes. 

Results: Consider improving the readability of this section. Consider respecting the stats hierarchy , first main and interactions effects, if has significant interaction, post hocs, if don’t, just the main effects.

As an example, your paragraph is (at least to primary and secondary outcomes):

"The TFA was affected by time (main effect: P < ..., EF = …) and group (main effect: P < ..., EF = …). Further, a significant interaction was observed (main effect time x group: P < ..., EF = …). In both groups, TFA increased as a function of time. Finally, significant differences between groups were observed at pretest (P < ..., EF = …) and posttest (P < ...) (Figure table xxxx)".

Discussion: first paragraph, please consider rewritten this paragraph in basis on primary and secondary outcomes (defined in the final of introduction). And after you re-analyze the results, persisting these negative results, please state clearly that (in terms of muscle mass, the RT precludes a gain on it, in comparison to control).

Discussion: One para addressing some potential applications of your findings can be useful for patients and health professionals.

Line 341 – were not

Lines 339-343 – Here, I see a huge problem of this experiment due to weak control of physical activity levels to both groups. Please, consider carefully discuss on possible deleterious effect of strength training. We know that any disruption to hormones related to catabolism and anabolism, like insulin produced in the pancreas, may also affect these processes and the overall metabolism. And, make clear that this rationale is speculative (as such your hypothesis of higher physical activity in which is less probable, because if it was true, the control group needed to do a very high training load to impact significantly in comparison to RT group)

Conclusion: as in previous parts, consider rewritten concluding strictly what you found, and considering future better controlled RCT’s to confirm these findings, and particularly, trying to find the optimal exposure-response of RT for individuals with pancreatic cancer.

Line 356 – occurs instead of happens

We look forward to receiving your revised manuscript.

Kind regards,

Leonardo A. Peyré-Tartaruga, Ph.D.

Academic Editor

PLOS ONE

Journal Requirements:

Reviewers' comments:

Reviewer's Responses to Questions

**Comments to the Author**

1. Is the manuscript technically sound, and do the data support the conclusions?

Reviewer #1: Partly

Reviewer #2: Yes

Reviewer #3: Partly

2. Has the statistical analysis been performed appropriately and rigorously? 

Reviewer #1: I Don't Know

Reviewer #2: Yes

Reviewer #3: No

3. Have the authors made all data underlying the findings in their manuscript fully available?

Reviewer #1: Yes

Reviewer #2: No

Reviewer #3: Yes

4. Is the manuscript presented in an intelligible fashion and written in standard English?

Reviewer #1: Yes

Reviewer #2: Yes

Reviewer #3: Yes

5. Review Comments to the Author

Reviewer #1: The paper entitled "Impact of progressive resistance training on CT quantified muscle and adipose tissue compartments in pancreatic cancer patients" wirtten by Wochner and colleagues aims to investigate the effectsof progressive resistance training onmuscle and adipose tissue compartments and the effect of body composition on overall survival. to this purpose 65 patients with pancreatic cancer were recruited and randomly assigned to either a training intervention or a control group. The 6-month intervention consisted of resistance training 3 times a weekwhile control group underwent usual care. Beforeand after the 6 months muscle strenght and CT scans were assessed. CT scans evaluated total-fat area, visceral fatarea, subcutaneous fat area, intramuscolar fat ares, and visceral tu subcutaneous fat ratio, muscle area, muscle density and skeletal musle index. Auhtors did not find any significant effec of the resistance training on muscle and adipose tissue compartments. However significant correlations were found between muscle mass and strenght parameters. Auhtors concluded that there is an anabolic potential with pancreatic cancer and progressive resistance training may be a promising tool that helps pancreatic cancer patients to maintain theri muscle mass and avoid muscle depletion.

Although the article is well and clearly written I personally have some concer:

1) How authors ensure the execution of the home-based training program? How the activity of the home-based group was monitored and recorded? Are authors sure that individuals included int he study attendend all the scheduled training sessions?

2) Many individuals were lost in the post-intervention assessment, resulting in 19 subjects in the intervention group and only 9 subjects in the control group. Don't you think that the lack of sttistical significant might be due to the great difference in the sample size of the two groups?

3) Have you thought to add a whitin group analysis?

4) Did you try to split the intervention group in RT1 and RT2 and check for possible differences?

5) How the intensity of the exercise was monitored in the home-based group? Is the RPE the right method to use to set the intensity of a resistance trainng intervention?

6) With an intervention of 2 times a week was the minimum amount of physical activity suggested by ACSM's guidelines for cancer patients achieved?

7) Authors decided to include in the resistance training major muscles for upper and lower estremities. Do authors have measured muscle mass of lower and upper limbs too? As far as I understood authors measured adipose tissue and muscle mass for:M. erector spinae, M. psoas major, M. rectus abdominis, M. obliquus internusabdominis, M. obliquus externus abdominis, M. transversus abdominis, M. quadratus, lumborum, M. latissimus dorsi, were those muscles directly involved in the resistance training?

8) Did authors performed some measure of muscle mass and adipose tissue in the trained limbs?

Reviewer #2: GENERAL COMMENTS:

This is a fine paper examining an important topic related to the effects of progressive resistance training on muscle and adipose tissue compartments and the effect of body composition on overall survival in pancreatic cancer patients. The topic of the study is original, and both the study design and the results presentation are sound. However, basic editing is needed and some basic questions require clarification. I have listed below specific comments to the authors for reference.

SPECIFIC COMMENTS:

Title:

I suggest the authors to include in the title the type of the study (i.e., Randomized Clinical Trial.

Abstract:

I suggest the authors to describe in the purpose that the exercise group was compared with usual care or control group.

Methods:

Which block size was used in the randomization process?

Results:

RT1 and RT2 presented similar baseline values for all outcomes? Is a limitation of the study pool the RT1 and RT2 data? Please, explain in more details.

Only intention-to-treat analysis was performed? Per protocol was analyzed? How the missing values were imputed in the analysis?

Is it possible to present the effect size for all outcomes?

The authors sharing the data in a public repository?

Reviewer #3: The manuscript entitled ‘Impact of progressive resistance training on CT quantified muscle and adipose tissue compartments in pancreatic cancer patients’ with the aim to investigate whether the intervention group showed a different course of body composition than the control group and to investigate whether there are any predictive factors in body composition that influence the survival of patients with pancreatic cancer.

The manuscript can be further improved based on the following comments.

Materials and Methods

Study population

Page 5 Line 111, the person who prepared the randomization block with allocation list and concealment to be stated. Likewise personnel involved in the recruitment and assessment. The impossible of blinding to be stated.

The exclusion criteria to be stated.

Statistical analysis

Page 9 Line 201=202, intent to treat basis not clear. Was there separate analysis for this? More information to be provided.

Page 9 Line 204-205, t-tests to be written as t-test (singular).

Page 9 Line 213, 1 or 2 tailed test to be stated in the sample size calculation.

Page 9 Line 215, the outcome variable to be clearly stated.

Results

The usage of n or N to be standardized throughout the manuscript.

Page 9 Line 224-225, sd to be stated apart from mean.

Page 11 Table 1, for ‘All patients, smoking (non smoker 83,0)’ the figure to be replaced with 75.5. Title is too short.

Page 13 & 14 Table 2 & 3, the word mean to be added to the adjusted difference. 95CI to be written as 95% CI. The word diff to be omitted and to be replaced with symbol * and denoted. Statistical test ANCOVA to be denoted in the table footnote.

Page 15 Line 283, correlation range values to be stated.

Page 16 Table 4, decimal point for the correlation value to be reduced (see also Line 277-283). Likewise with the p values and to be standardized.

Table 5 & 6. the lower and upper 95% CI to be presented in the following format

95% CI

Lower Upper

S1 Table 7 and S1 Table 8, summary findings to be stated in the results section.

Effect size could be presented.

Discussion

Page 28 Line 440, 'supplementary materials' to be replaced with S1 Table 7 and S1 Table 8.

Apart from the limitation discussed, what was the post hoc study power based on the final sample size (although it may be controversial to discuss) including the statistical analysis.

The source of funding for the study to be stated in the manuscript.

6. PLOS authors have the option to publish the peer review history of their article (what does this mean?). If published, this will include your full peer review and any attached files.

Reviewer #1: No

Reviewer #2: **Yes: **Stephanie Santana Pinto

Reviewer #3: No

---

## [Author Response · Author response to Decision Letter 0]

15 Sep 2020

PONE-D-20-06946

Impact of progressive resistance training on CT quantified muscle and adipose tissue compartments in pancreatic cancer patients

PLOS ONE

Thank you for carefully reviewing our manuscript and providing us with very helpful and constructive comments. We revised the manuscript accordingly.

Academic editor:

Please, send the dataset to specific supplementary material server managed by the library service of the German Cancer Research Center.

REPLY: We sent the minimal data set to the supplementary material server of the library service of the German Cancer Research Center. The patients’ informed consent did not include public data sharing. Due to protection of private data, the data set is intended only for the review process for the editors and reviewers and needs to be handled confidentially.

Below the link to the data set:

http://suppl.dkfz.de/KS/Support_minimal_Data_set_24_07_2020.xlsx

Thank you for your confidentiality.

Abstract, line 29: which hypothesis was tested? (please state it).

REPLY: We rephrased the sentence to make it clearer which was our primary aim within this explorative analysis.

Abstract, lines 29-31: Please put the primary outcome first to indicate the hierarchy. Please also clearly state what was the primary outcome.

REPLY: We added the hierarchy of the analyses. As all analyses presented in this manuscript were based on secondary endpoints of the randomized SUPPORT-trial, we prefer not to use the term primary outcome. We defined a primary and secondary aim for the analyses presented here and revised and standardized the terminology throughout the manuscript. 

Abstract, results: Please add effect sizes and not just p-values. Focus on group contrasts in the reporting (please see the PREPARE Trial guide for guidance). And, consider including ES and p-values in the non-differences found.

REPLY: Thank you for the indication. We added also effect sizes.

Abstract, end: Please add clinical-trial registration-info at the end of the abstract. Because it seems as if the trial was retrospectively registered (registration after inclusion of the first participant) add “retrospectively registered” after the trial registration number. Please state clearly in the manuscript if the primary outcome was pre-defined (defined before inclusion of the first participant).

REPLY: We added the clinical-trial registration info at the end of the abstract. The study record was submitted on October 30th 2013 and the recruitment of the study started in December 2013.

Secondary outcomes defined before the start of the study included body weight and body composition with mentioning the routinely performed CT scans as information source in the study protocol. 

Hypotheses: Consider including objective hypotheses, not just diff or not, but y higher than x condition style.

REPLY: We rephrased the sentence and added a specific hypothesis on page 4.

Was it possible to account for confounding factors across groups, such as the effect of pharmacological doses? 

REPLY: Unfortunately, we were not able to assess data on pharmacological doses or chemotherapy completion rates within our study. Not all patients received their chemotherapy treatment at the National Center for Tumor Diseases in Heidelberg. Patients who lived further away received their treatment at an oncologist close to their homes.

Please remove statistical tests for baseline differences. CONSORT advise against this. Please see http://www.consort-statement.org/Media/Default/Downloads/CONSORT%202010%20Explanation%20and%20Elaboration%20Document-BMJ.pdf page 17.

REPLY: Thank you for the indication. We removed the statistical tests for baseline differences.

Results and Stats: please report 95CI of all variables and effect sizes. 

REPLY: We added 95CI of all variables and effect sizes. We used partial omega squared instead of eta squared due to correcting the bias and removing sampling error influences and due to small sample size.

Results: Consider improving the readability of this section. Consider respecting the stats hierarchy, first main and interactions effects, if has significant interaction, post hocs, if don’t, just the main effects.

As an example, your paragraph is (at least to primary and secondary outcomes):

"The TFA was affected by time (main effect: P < ..., EF = …) and group (main effect: P < ..., EF = …). Further, a significant interaction was observed (main effect time x group: P < ..., EF = …). In both groups, TFA increased as a function of time. Finally, significant differences between groups were observed at pretest (P < ..., EF = …) and posttest (P < ...) (Figure table xxxx)".

REPLY: Since we performed analyses of covariance to assess the differences in body composition between the groups from pre- to post-intervention we didn’t obtain the main effect results (as analyzed in the mixed model analyses). Therefore, we could not report the main effects of time and group. 

Discussion: first paragraph, please consider rewritten this paragraph in basis on primary and secondary outcomes (defined in the final of introduction). And after you re-analyze the results, persisting these negative results, please state clearly that (in terms of muscle mass, the RT precludes a gain on it, in comparison to control).

REPLY: Thank you for the indication. We rephrased the first paragraph of the discussion on page 21 to be in line with the definition of the hypothesis in the introduction and stated, that we couldn’t confirm the hypothesis (primary aim) in this study.

Discussion: One para addressing some potential applications of your findings can be useful for patients and health professionals.

REPLY: We added potential applications in the conclusion part of the discussion on page 26: Identify patients, who could profit from training, monitoring of training and also risk stratification of patients.

Line 341 – were not

REPLY: Thank you for the indication. We corrected the terminology.

Lines 339-343 – Here, I see a huge problem of this experiment due to weak control of physical activity levels to both groups. Please, consider carefully discuss on possible deleterious effect of strength training. We know that any disruption to hormones related to catabolism and anabolism, like insulin produced in the pancreas, may also affect these processes and the overall metabolism. And, make clear that this rationale is speculative (as such your hypothesis of higher physical activity in which is less probable, because if it was true, the control group needed to do a very high training load to impact significantly in comparison to RT group)

REPLY: We agree that the weak control of physical activity levels is a huge limitation of this study. Due to small sample size individuals could have a huge impact and we don’t have a verifiable explanation for the increase in muscle mass in CON. We emphasized that our explanations are speculative and added changes to anabolic hormones such as insulin as possible explanation on page 21 and 22. 

Conclusion: as in previous parts, consider rewritten concluding strictly what you found, and considering future better controlled RCT’s to confirm these findings, and particularly, trying to find the optimal exposure-response of RT for individuals with pancreatic cancer.

REPLY: We rephrased the conclusion on page 25 and 26 to be more in line with the hypothesis introduced in the introduction.

Line 356 – occurs instead of happens

REPLY: Thank you for the indication. We corrected the terminology.

Reviewers' comments to the Author:

Reviewer #1: The paper entitled "Impact of progressive resistance training on CT quantified muscle and adipose tissue compartments in pancreatic cancer patients" written by Wochner and colleagues aims to investigate the effects of progressive resistance training on muscle and adipose tissue compartments and the effect of body composition on overall survival. to this purpose 65 patients with pancreatic cancer were recruited and randomly assigned to either a training intervention or a control group. The 6-month intervention consisted of resistance training 3 times a week while control group underwent usual care. Before and after the 6 months muscle strength and CT scans were assessed. CT scans evaluated total-fat area, visceral fat area, subcutaneous fat area, intramuscular fat areas, and visceral to subcutaneous fat ratio, muscle area, muscle density and skeletal muscle index. Authors did not find any significant effect of the resistance training on muscle and adipose tissue compartments. However significant correlations were found between muscle mass and strength parameters. Authors concluded that there is an anabolic potential with pancreatic cancer and progressive resistance training may be a promising tool that helps pancreatic cancer patients to maintain their muscle mass and avoid muscle depletion.

Although the article is well and clearly written I personally have some concerns:

1) How authors ensure the execution of the home-based training program? How the activity of the home-based group was monitored and recorded? Are authors sure that individuals included in the study attended all the scheduled training sessions?

REPLY: We included further information in the methods section on page 6 about recording the training. The monitoring of the home-based group was done by weekly phone calls of the exercise therapist.

Further, we also included information on the training adherence rate of the patients in the intervention group in the results section on page 11, as this information was previously missing. Thank you for the good indication.

2) Many individuals were lost in the post-intervention assessment, resulting in 19 subjects in the intervention group and only 9 subjects in the control group. Don't you think that the lack of statistical significance might be due to the great difference in the sample size of the two groups?

REPLY: We agree with your statement. The unbalanced group size of the intervention group and the control group could be an influencing factor. We stated this already in our limitations on page 25. Due to ionizing radiation of CT no additional CT scans were performed and only CT scans of clinical routine indication were used. Therefore, we needed to exclude patients who didn’t have an eligible CT-scan.

3) Have you thought to add a within group analysis?

REPLY: In addition to the ANCOVA (main analysis) we also performed paired t-test for the changes from T0 to T2 for each group. The results are also presented in Table 2 and 3.

4) Did you try to split the intervention group in RT1 and RT2 and check for possible differences?

REPLY: As there were only n= 4 patients in the supervised resistance training group (RT1) and n= 15 patients in the home-based resistance training group (RT2) with no differences between the groups, we decided to combine both training groups to a pooled resistance training group (RT) for the analysis.

5) How the intensity of the exercise was monitored in the home-based group? Is the RPE the right method to use to set the intensity of a resistance training intervention?

REPLY: The intensity in the home-based group was monitored with the Borg Scale of Perceived Exertion by weekly phone calls. We agree that monitoring the intensity with guidelines of, for example, 60-80% 1RM under supervision is better than training at home with intensities monitored with the RPE. However, we believe that prior to the fact that patients exercise alone at home during chemotherapy, monitoring intensity with the RPE is feasible and justifiable.

6) With an intervention of 2 times a week was the minimum amount of physical activity suggested by ACSM's guidelines for cancer patients achieved?

REPLY: The updated ACSM guidelines for exercise for cancer patients from 2019 recommend to perform resistance training 2 times a week with 2 sets with 8-15 repetitions with 60% of the one repetition maximum for the major muscle groups. Based on this suggestion, we meet the requirements to generate a positive effect on the quality of life, fatigue and physical function. At present, pancreatic cancer patients follow the general ACSM recommendations for exercise for cancer patients. 

7) Authors decided to include in the resistance training major muscles for upper and lower extremities. Do authors have measured muscle mass of lower and upper limbs too? As far as I understood authors measured adipose tissue and muscle mass for: M. erector spinae, M. psoas major, M. rectus abdominis, M. obliquus internus abdominis, M. obliquus externus abdominis, M. transversus abdominis, M. quadratus, lumborum, M. latissimus dorsi, were those muscles directly involved in the resistance training?

REPLY: Unfortunately, we didn’t assess muscle mass of the upper and lower limbs. The above stated muscles were not directly targeted in the resistance training, but were partially involved as overall stabilizing muscles during the exercise. For example, squats were used as exercise, at which the core muscles like M. erector spinae are used as stabilizing muscles.

There is evidence, that measurement of muscle tissue on a single slice image on vertebra L3 level shows a high correlation with whole body muscle mass:

Shen, W., M. Punyanitya, Z. Wang, D. Gallagher, M. P. St-Onge, J. Albu, S. B. Heymsfield, and S. Heshka. 2004. 'Total body skeletal muscle and adipose tissue volumes: estimation from a single abdominal cross-sectional image', J Appl Physiol (1985), 97: 2333-8

Therefore, we took the L3 SMI as a surrogate parameter for whole body muscle mass.

8) Did authors perform some measure of muscle mass and adipose tissue in the trained limbs?

REPLY: Unfortunately, we didn’t. We had already mentioned this as a limiting factor in the discussion on page 25. Imaging of extremities (arms and legs) is very rare in routine clinical context with patients with pancreatic cancer and not frequently performed. Therefore, no routine imaging of those body regions was available and we weren’t able to measure the muscle mass of the limbs directly. 

Please see also the answer above.

Reviewer #2: GENERAL COMMENTS:

This is a fine paper examining an important topic related to the effects of progressive resistance training on muscle and adipose tissue compartments and the effect of body composition on overall survival in pancreatic cancer patients. The topic of the study is original, and both the study design and the results presentation are sound. However, basic editing is needed and some basic questions require clarification. I have listed below specific comments to the authors for reference.

SPECIFIC COMMENTS:

Title: I suggest the authors to include in the title the type of the study (i.e., Randomized Clinical Trial.

REPLY: The analyses presented focus on secondary endpoints of a prospective randomized controlled trial. Thus, we tend to not mention the study type (randomized controlled trial) in the title.

Abstract: I suggest the authors to describe in the purpose that the exercise group was compared with usual care or control group.

REPLY: This concern was addressed in the methods section of the abstract on page 2 line 33,34. Due to the limiting word count we did not add the information also in the purpose. 

Methods: Which block size was used in the randomization process?

REPLY: A 2:1 block randomization with varying block sizes of 3 and 6 was used. We added the varying block sizes to the method section on page 6.

Results: RT1 and RT2 presented similar baseline values for all outcomes? Is a limitation of the study pool the RT1 and RT2 data? Please, explain in more details.

REPLY: As there were only n= 4 patients in the supervised resistance training group (RT1) and n= 15 patients in the home-based training group (RT2) with no differences between the groups, we decided to combine both training groups to a pooled resistance training group (RT) for the analysis.

We agree that the pooling could be a limitation, as we mentioned in the discussion of the manuscript on page 25.

Only intention-to-treat analysis was performed? Per protocol was analyzed? How the missing values were imputed in the analysis?

REPLY: The presented subsequent analysis on routine CT scans was intention-to-treat. There were no missing values imputed in the analysis, as missing values most likely occurred due to patients’ death or disease progression or due to non-existent CT scans or not evaluable, not suitable CT scans. 

Is it possible to present the effect size for all outcomes?

REPLY: Thank you for the indication. We added also effect sizes. We used partial omega squared instead of eta squared due to correcting the bias and removing sampling error influences and due to small sample size.

The authors sharing the data in a public repository?

REPLY: Yes, the minimal dataset was sent to specific supplementary material server managed by the library service of the German Cancer Research Center. However, due to the data protection regulations applicable here in the context of the study (the patients’ informed consent did not include public data sharing), the data will only be available for the review process for the editor and the reviewers and needs to be handled confidentially. See also answer to first comment of the editor.

Reviewer #3: The manuscript entitled ‘Impact of progressive resistance training on CT quantified muscle and adipose tissue compartments in pancreatic cancer patients’ with the aim to investigate whether the intervention group showed a different course of body composition than the control group and to investigate whether there are any predictive factors in body composition that influence the survival of patients with pancreatic cancer.

The manuscript can be further improved based on the following comments.

Materials and Methods

Study population

Page 5 Line 111, the person who prepared the randomization block with allocation list and concealment to be stated. Likewise personnel involved in the recruitment and assessment. The impossible of blinding to be stated.

REPLY: We added a sentence about blinding within the SUPPORT-study on page 6. The Randomization of a patient was done by an independent biometrician.

The exclusion criteria to be stated.

REPLY: We added the exclusion criteria to the paragraph of the study population within the Materials & Methods section on page 5.

Statistical analysis

Page 9 Line 201=202, intent to treat basis not clear. Was there separate analysis for this? More information to be provided.

REPLY: We added further information and rephrased the sentence on page 9 to make it more clear.

Page 9 Line 204-205, t-tests to be written as t-test (singular).

REPLY: We corrected the terminology as you recommended.

Page 9 Line 213, 1 or 2 tailed test to be stated in the sample size calculation.

REPLY: We used the 2-sided test. We have slightly reworded the sentence. But since this is an explorative analysis of CT scan data, we decided to delete this sentence with the sample size calculation. Please, see also the answer below.

Page 9 Line 215, the outcome variable to be clearly stated.

REPLY: The outcome variable of the sample size calculation for the SUPPORT study was the physical functioning subscale of the quality of Life questionnaire (EORTC-QLQ-C30). No further power calculation was performed for the explorative sub-analysis presented here. Therefore, we decided to delete the part with the description of the sample size calculation on page 9 and added the following sentence instead: “For the presented explorative analysis on routine CT scans no further power calculation was performed.”

Results

The usage of n or N to be standardized throughout the manuscript.

REPLY: Thank you for the indication. We have standardized the use of “n” in the manuscript.

Page 9 Line 224-225, sd to be stated apart from mean.

REPLY: Thank you for the indication. We added the SD.

Page 11 Table 1, for ‘All patients, smoking (non smoker 83,0)’ the figure to be replaced with 75.5. Title is too short.

REPLY: Thank you very much for the indication. We have corrected the mistake. Further, we extended the title of Table 1 with more information.

Page 13 & 14 Table 2 & 3, the word mean to be added to the adjusted difference. 95CI to be written as 95% CI. The word diff to be omitted and to be replaced with symbol * and denoted. Statistical test ANCOVA to be denoted in the table footnote.

REPLY: Thank you for the indication. We adjusted the tables 2 and 3 on page 14 and 15.

Page 15 Line 283, correlation range values to be stated.

REPLY: We added the correlation range values.

Page 16 Table 4, decimal point for the correlation value to be reduced (see also Line 277-283). Likewise with the p values and to be standardized.

REPLY: We reduced the decimal point for the correlation coefficient to 2 and p-values were standardized to 3 decimal points in Table 4.

Table 5 & 6. the lower and upper 95% CI to be presented in the following format

95% CI

Lower Upper

REPLY: Thank you for the indication. We adjusted the tables 5 and 6.

S1 Table 7 and S1 Table 8, summary findings to be stated in the results section.

REPLY: A paragraph about S1 and S2 was added in the results section on page 16.

Effect size could be presented.

REPLY: We added effect sizes. We used partial omega squared instead of eta squared due to correcting the bias and removing sampling error influences and due to small sample size.

Discussion

Page 28 Line 440, 'supplementary materials' to be replaced with S1 Table 7 and S1 Table 8.

REPLY: Thank you for the indication. We adjusted the wording.

Apart from the limitation discussed, what was the post hoc study power based on the final sample size (although it may be controversial to discuss) including the statistical analysis.

REPLY: Since this is an explorative analysis of CT scan data, no further power calculation was performed.

The source of funding for the study to be stated in the manuscript.

REPLY: We followed this comment by adding the source of funding for the study at the end of the manuscript on page 27.

---

## [Decision Letter · Decision Letter 1]

10 Nov 2020

Impact of progressive resistance training on CT quantified muscle and adipose tissue compartments in pancreatic cancer patients

PONE-D-20-06946R1

Dear Dr. Steindorf,

We’re pleased to inform you that your manuscript has been judged scientifically suitable for publication and will be formally accepted for publication once it meets all outstanding technical requirements.

Kind regards,

Leonardo A. Peyré-Tartaruga, Ph.D.

Academic Editor

PLOS ONE

Additional Editor Comments (optional):

Reviewers' comments:

Reviewer's Responses to Questions

**Comments to the Author**

1. If the authors have adequately addressed your comments raised in a previous round of review and you feel that this manuscript is now acceptable for publication, you may indicate that here to bypass the “Comments to the Author” section, enter your conflict of interest statement in the “Confidential to Editor” section, and submit your "Accept" recommendation.

Reviewer #1: All comments have been addressed

Reviewer #3: (No Response)

2. Is the manuscript technically sound, and do the data support the conclusions?

Reviewer #1: Yes

Reviewer #3: Partly

3. Has the statistical analysis been performed appropriately and rigorously? 

Reviewer #1: Yes

Reviewer #3: No

4. Have the authors made all data underlying the findings in their manuscript fully available?

Reviewer #1: Yes

Reviewer #3: Yes

5. Is the manuscript presented in an intelligible fashion and written in standard English?

Reviewer #1: Yes

Reviewer #3: Yes

6. Review Comments to the Author

Reviewer #1: Authors have addressed all the comments of this reviewer.

This reviewer does not have any other suggestion.

Reviewer #3: The authors have put in effort to address the comments.

Minor comment.

Please ensure the partial omega squared symbol is correctly inserted/copied into the manuscript and comments by academic editor on statistical presentation are addressed.

7. PLOS authors have the option to publish the peer review history of their article (what does this mean?). If published, this will include your full peer review and any attached files.

Reviewer #1: No

Reviewer #3: No

---

## [Editor Report · Acceptance letter]

16 Nov 2020

PONE-D-20-06946R1 

Impact of progressive resistance training on CT quantified muscle and adipose tissue compartments in pancreatic cancer patients 

Dear Dr. Steindorf:

I'm pleased to inform you that your manuscript has been deemed suitable for publication in PLOS ONE. Congratulations! Your manuscript is now with our production department. 

Kind regards, 

on behalf of

Professor Leonardo A. Peyré-Tartaruga 

Academic Editor

PLOS ONE